# Myofibril diameter is set by a finely tuned mechanism of protein oligomerization in *Drosophila*

Nicanor González-Morales, Yu Shu Xiao, Matthew Aaron Schilling, Océane Marescal, Kuo An Liao, Frieder Schöck*

Department of Biology, McGill University, Montreal, Canada

**Abstract** Myofibrils are huge cytoskeletal assemblies embedded in the cytosol of muscle cells. They consist of arrays of sarcomeres, the smallest contractile unit of muscles. Within a muscle type, myofibril diameter is highly invariant and contributes to its physiological properties, yet little is known about the underlying mechanisms setting myofibril diameter. Here we show that the PDZ and LIM domain protein Zasp, a structural component of Z-discs, mediates Z-disc and thereby myofibril growth through protein oligomerization. Oligomerization is induced by an interaction of its ZM domain with LIM domains. Oligomerization is terminated upon upregulation of shorter Zasp isoforms which lack LIM domains at later developmental stages. The balance between these two isoforms, which we call growing and blocking isoforms sets the stereotyped diameter of myofibrils. If blocking isoforms dominate, myofibrils become smaller. If growing isoforms dominate, myofibrils and Z-discs enlarge, eventually resulting in large pathological aggregates that disrupt muscle function.

## Introduction

Inside cells, proteins are assembled into complex functional units. The correct assembly of these units is crucial for their function (*Marsh and Teichmann, 2015*). Myofibrils are highly organized assemblies of cytoskeletal proteins forming an array of sarcomeres that are embedded in the cytosol of myotubes and mediate contractility (*Huxley and Niedergerke, 1954b*; *Huxley and Hanson, 1954a*; *Huxley, 2004*; *Lemke and Schnorrer, 2017*). Sarcomeres are composed of actin-containing thin filaments and myosin-containing thick filaments arranged into antiparallel cables. Thin filaments are anchored to a large multiprotein complex called the Z-disc (*Luther, 2009*), and thick filaments are anchored to another large multiprotein complex called the M-line (*Agarkova and Perriard, 2005*). Anchoring of myofibrils to the exoskeleton provides the mechanical tension that aligns sarcomeres into myofibrils and coordinates their development (*Weitkunat et al., 2017*; *Weitkunat et al., 2014*). Once aligned, sarcomeres grow to their final size (*Lemke and Schnorrer, 2017*). Electron and confocal microscopy studies showed that sarcomeres form initially from small structures called Z-bodies that grow eventually into mature Z-discs to which thin filaments are anchored (*Loison et al., 2018*; *Orfanos et al., 2015*; *Reedy and Beall, 1993*; *Shafiq, 1963*; *Sparrow and Schöck, 2009*). The size of the Z-disc therefore sets the diameter of the myofibril (*Agarkova and Perriard, 2005*; *Luther, 2009*). While mechanisms have been proposed that set the length of sarcomeres (*Fernandes and Schöck, 2014*; *Gokhin and Fowler, 2013*; *Tskhovrebova and Trinick, 2017*), Z-disc growth and growth termination is poorly understood.

A hallmark of genetically caused myopathies is the appearance of large aggregates composed mainly of Z-disc proteins (*Kley et al., 2016*; *Maerkens et al., 2016*). Interestingly, many myopathy-associated mutations encode Z-disc proteins. Mutations in any of the four α-actinin genes or in Zasp and other Alp/Enigma family genes in humans cause myopathies characterized by the presence of

*For correspondence:
frieder.schoeck@mcgill.ca

Competing interests: The authors declare that no competing interests exist.

**eLife digest** Muscles are made up of many long muscle fibers, each containing thousands of cylindrical segments called sarcomeres. When animals move, proteins in the sarcomere move past each other, shortening the muscles. Inside each muscle, all sarcomeres have the same length and diameter. The protein titin controls the length of each sarcomere, but it was unknown what controls the diameter.

At the end of each sarcomere is a structure called the Z-disc that is composed of many muscle proteins. Mutations in Z-disc proteins are often involved in diseases called myopathies, where muscle structure breaks down. As the size of the Z-disc determines sarcomere diameter, improper regulation of sarcomere diameter could contribute to myopathies. One Z-disc protein called Zasp is a candidate for controlling diameter and can have many different forms in the same cells. Zasp has a similar role in most animals including humans, mice and flies.

González-Morales et al. investigated Zasp in the muscles of the fruit fly, *Drosophila melanogaster*. Gene editing was used to vary the amounts of different forms of Zasp inside the muscles. The results revealed two types of Zasp, those that make sarcomeres wider, and those that limit growth. Reducing the second type of Zasp resulted in bigger Z-discs and in muscle aggregates similar to the ones seen in patients with certain myopathies.

This study reveals a mechanism for coordinating the development of muscle. It also reveals the likely cause of certain myopathies and suggests a possible target for future treatment through regulation of Zasp proteins.

large aggregates (*Murphy and Young, 2015*; *Selcen and Engel, 2005*). The aggregation phenotype is conserved among animals: fruit flies with mutations in myopathy-related genes also develop Z-disc aggregates (*González-Morales et al., 2017*).

α-Actinin and Z-disc Alternatively Spliced Protein (Zasp) are conserved proteins that coordinate Z-disc formation (*Faulkner et al., 1999*; *Katzemich et al., 2013*; *Murphy and Young, 2015*). α-Actinin forms a rod-shaped antiparallel homodimer at the Z-disc, where it crosslinks and serves as an attachment point for actin filaments (*Djinović-Carugo et al., 1999*; *Luther, 2009*; *Murphy and Young, 2015*; *Ribeiro et al., 2014*; *Rusu et al., 2017*; *Takahashi and Hattori, 1989*). Zasp and other members of the Alp/Enigma family of proteins are scaffolding proteins with an α-actinin-binding PDZ domain (InterPro: IPR001478), an uncharacterized Zasp Motif (ZM; InterPro: IPR031847 and IPR006643) domain, and one to four protein-protein interaction LIM domains (InterPro: IPR001781) (*Finn et al., 2017*; *Klaavuniemi et al., 2004*; *Liao et al., 2016*). Zasp and α-actinin proteins are present at the earliest stages of Z-disc formation, and are required for Z-disc assembly (*Dabiri et al., 1997*; *Katzemich et al., 2013*).

Vertebrates have seven Alp/Enigma genes, each encoding several isoforms. The *Drosophila* genome has three Zasp genes, *Zasp52*, *Zasp66*, and *Zasp67*, which encode 21, 12, and 4 isoforms, respectively (*Gramates et al., 2017*). Zasp66 and Zasp67 are duplications of Zasp52 and resemble the smallest isoforms of Zasp52 (*González-Morales et al., 2019*). The number of isoform variants and genes adds an additional layer of complexity and regulation to sarcomere formation.

We used *Drosophila* indirect flight muscles and asked if the mechanism that controls Z-disc size relates to the pathological aggregation behavior known for Z-disc-related myopathies. We show that accumulation of multivalent Zasp growing isoforms (with multiple LIM domains) causes Z-disc growth, whereas upregulation of monovalent blocking isoforms at later developmental time points terminates Z-disc growth. An imbalance of growing and blocking Zasp isoforms results either in aggregate formation, enlarged Z-disc size or reduced Z-disc size. We propose that this mechanism has wide implications for diseases caused by aggregate formation.

## Results

### Aggregate formation upon Zasp overexpression

Dominant Zasp mutations cause aggregate formation in human myopathies (*Selcen and Engel, 2005*). To examine if this holds true in *Drosophila*, we overexpressed a full-length Zasp52-PR

transgene consisting of a PDZ, a ZM, and 4 LIM domains in the indirect flight muscles (IFM) of adult

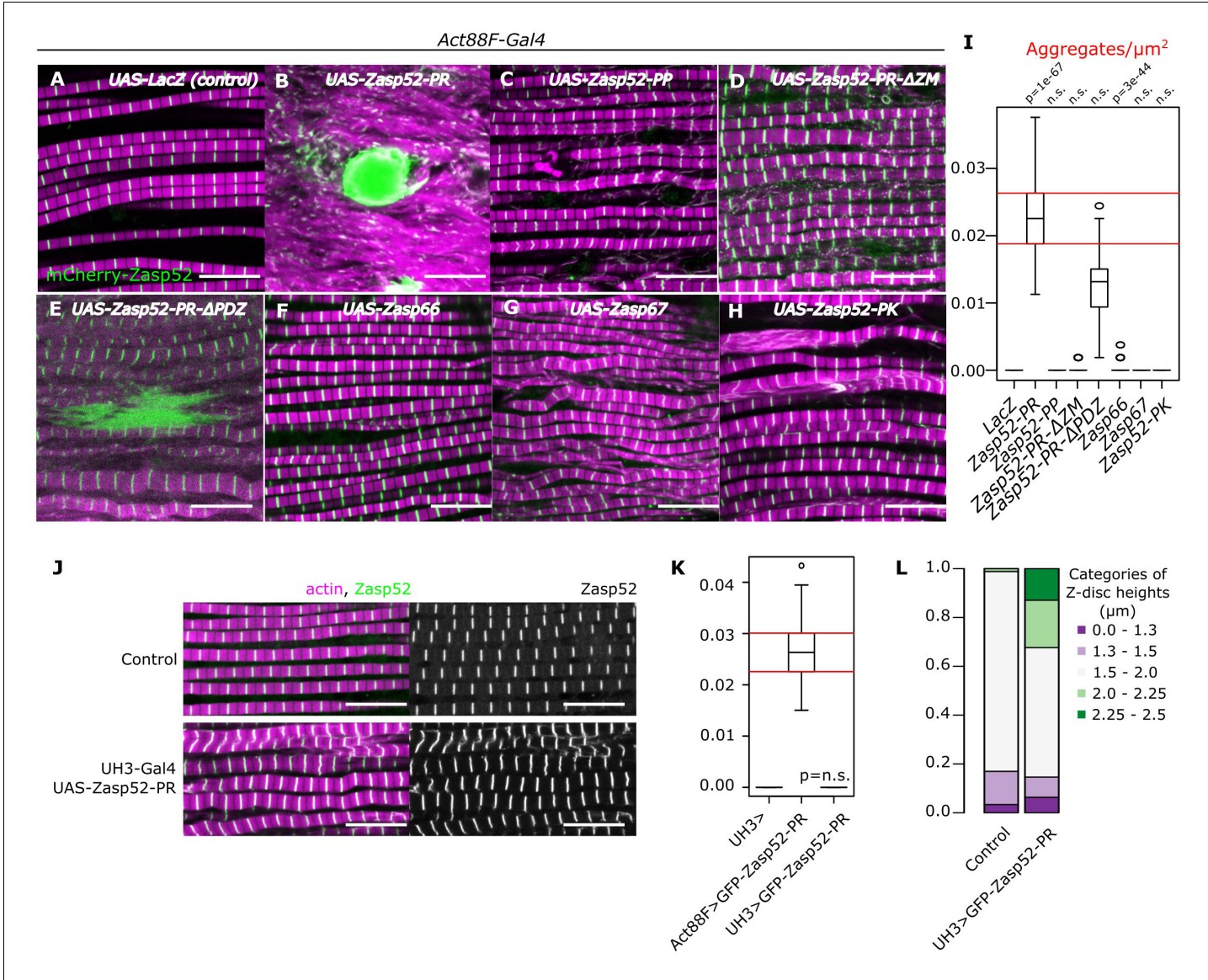

**Figure 1.** Aggregate formation upon Zasp overexpression. (A–H) Confocal images of IFM in control or Zasp overexpression conditions using Act88F-Gal4; actin filaments are marked in magenta and Z-discs in green. (A) In control flies, sarcomeres are regular with Z-discs located in the center of the actin signal. (B) Upon GFP-Zasp52-PR overexpression sarcomere structure is severely affected and big Zasp aggregates appear. (C) The overexpression of Flag-Zasp52-PP, an isoform lacking all LIM domains, has a modest sarcomere phenotype and aggregation is not observed. (D) Overexpression of a mutated form of Zasp52-PR with a deletion in the ZM domain does not form aggregates. (E) Overexpression of a mutated form of Zasp52-PR carrying a deletion in the PDZ domain causes small aggregates and sarcomere phenotypes. (F and G) The overexpression of the other Zasp paralogs, Zasp66 or Zasp67, does not cause aggregates. (H) Overexpression of GFP-Zasp52-PK, a smaller isoform of Zasp52 lacking LIM2, 3, and 4 domains affects sarcomere structure, but aggregation is infrequent. (I) Estimation of Z-disc aggregates in all overexpression conditions, n = 10 muscle fibers. (J) Confocal images of IFM in control or in mild overexpression of GFP-Zasp52-PR using UH3-Gal4. (K) Estimation of Z-disc aggregates in mild overexpression of GFP-Zasp52-PR. (L) Frequency plot of Z-disc sizes. Z-discs are bigger when GFP-Zasp52-PR is overexpressed using UH3-Gal4. Scale bars, 10 μm. p-Values in panel I and K were calculated using Welch's two-sample t-test.

The online version of this article includes the following source data and figure supplement(s) for figure 1:

**Source data 1.** Aggregate frequency estimates in Zasp overexpressions.
**Figure supplement 1.** Differences in Zasp isoforms.
**Figure supplement 2.** Zasp-mediated aggregation depends only partially on α-actinin.

flies, which causes formation of large aggregates (*Figure 1A,B,I*, *Figure 1—figure supplement 1*). To determine which domain is responsible for aggregation, we deleted them individually: the absence of LIM domains (Zasp52-PP) and ZM domain (Zasp52-PRΔZM) abolished aggregate formation, whereas removal of the PDZ domain only slightly reduced the number of aggregates (Zasp52-PRΔPDZ, *Figure 1C–E,I*). The PDZ domain is required for binding α-actinin, a crucial crosslinker of actin filaments at the Z-disc (*Katzemich et al., 2013*; *Liao et al., 2016*). Reducing α-actinin levels by RNAi in Zasp52-PR overexpression reduces aggregate formation to a similar level (*Figure 1—figure supplement 2*), indicating that α-actinin increases aggregate number, but is not required for their initial formation. Expression of the paralogous genes Zasp66 and Zasp67 (*González-Morales et al., 2019*), which lack LIM domains like Zasp52-PP, also does not result in aggregate formation (*Figure 1F,G and I*). Finally, we tested if multiple LIM domains are required for aggregate formation by overexpressing Zasp52-PK consisting of PDZ, ZM and 1 LIM domain. No aggregates form (*Figure 1H and I*) indicating that the ZM domain and multiple LIM domains are required for aggregate formation.

To test if aggregates formed as a result of Z-disc overgrowth, we tested the overexpression of Zasp52-PR using UH3-Gal4, a driver line that has the same temporal and spatial expression pattern as Act88-Gal4 but is expressed at a much lower level. To better capture the size variation, we measured the Z-disc height that corresponds to the disc diameter and categorized them into specific size categories. In this condition, Z-discs are bigger than the control, but aggregates are not present (*Figure 1J–L*), indicating that Z-disc growth and pathological aggregation correlate with the amount of Zasp52 protein.

## Domains required for Zasp self-interaction

Next, we employed the yeast two-hybrid system (Y2H) to determine if Zasp forms oligomers by interacting with itself. Zasp52-PK and Zasp52-PE, another full-length isoform, can interact with each other, whereas controls do not interact (*Figure 2A*). Full-length isoforms of Zasp66 and Zasp67 (Zasp66-PK and Zasp67-PD) can also interact with Zasp52-PK and Zasp52-PE (*Figure 2A*). These data suggest that Zasp proteins can homo- and heterodimerize.

We confirmed these results by co-immunoprecipitation of Flag-tagged Zasp52 with GFP-tagged Zasp52 from thorax muscle extracts (*Figure 2—figure supplement 1A*). Furthermore, bacterially purified His-Zasp52-PK-Flag dimerized in vitro in a chemical crosslinking assay (*Figure 2—figure supplement 1B*).

To identify the domains involved in homo- and heterodimerization, we tested the interaction of Zasp52-PK, Zasp66 and Zasp67 with each individual domain of Zasp52. Zasp52-PK interacts with the ZM domain and LIM domains, whereas Zasp66 and Zasp67 only interact with LIM domains (*Figure 2B*). This suggests that dimerization is mediated by a ZM-LIM domain interaction, and because Zasp66 and Zasp67 lack LIM domains, they cannot interact with the ZM domain (*Figure 2C*). To confirm this hypothesis, we next tested the interaction of LIM2A, LIM2B, LIM3 and Zasp52-PK with a series of Zasp66 deletion constructs. As soon as the ZM domain of Zasp66 is deleted, the interaction with LIM domains is abolished (*Figure 2D*). Finally, we tested Zasp66-PH, a ZM-only isoform of Zasp66 for interaction with the individual domains of Zasp52. Zasp66-PH can only interact with LIM domains of Zasp52 (*Figure 2E*). Thus, LIM-ZM binding mediates homo- and heterodimerization between Zasp proteins.

## Bimolecular fluorescence complementation reveals Zasp interaction at the Z-disc

Next, we investigated if LIM-ZM interaction observed in a heterologous system also occurs in vivo at the Z-disc. We hypothesized that an endogenous GFP-tagged Zasp66 will be recruited to the Z-discs by Zasp52. We analyzed two Zasp52 mutants that differentially affect sarcomere structure: *Zasp52$^{MI02988}$* disrupts N-terminal isoforms and only partially affects the last three LIM domains, whereas *Zasp52$^{MI00979}$* introduces a stop codon before the last three LIM domains (*Figure 3—figure supplement 1A*) (*Liao et al., 2016*). Zasp66-GFP fluorescence is mildly reduced in *Zasp52$^{MI02988}$* mutants (*Figure 3A,B and D*). In contrast, Zasp66-GFP fluorescence is strongly reduced in *Zasp52$^{MI00979}$* mutants (*Figure 3C and D*), suggesting that Zasp52 recruits other Zasp proteins

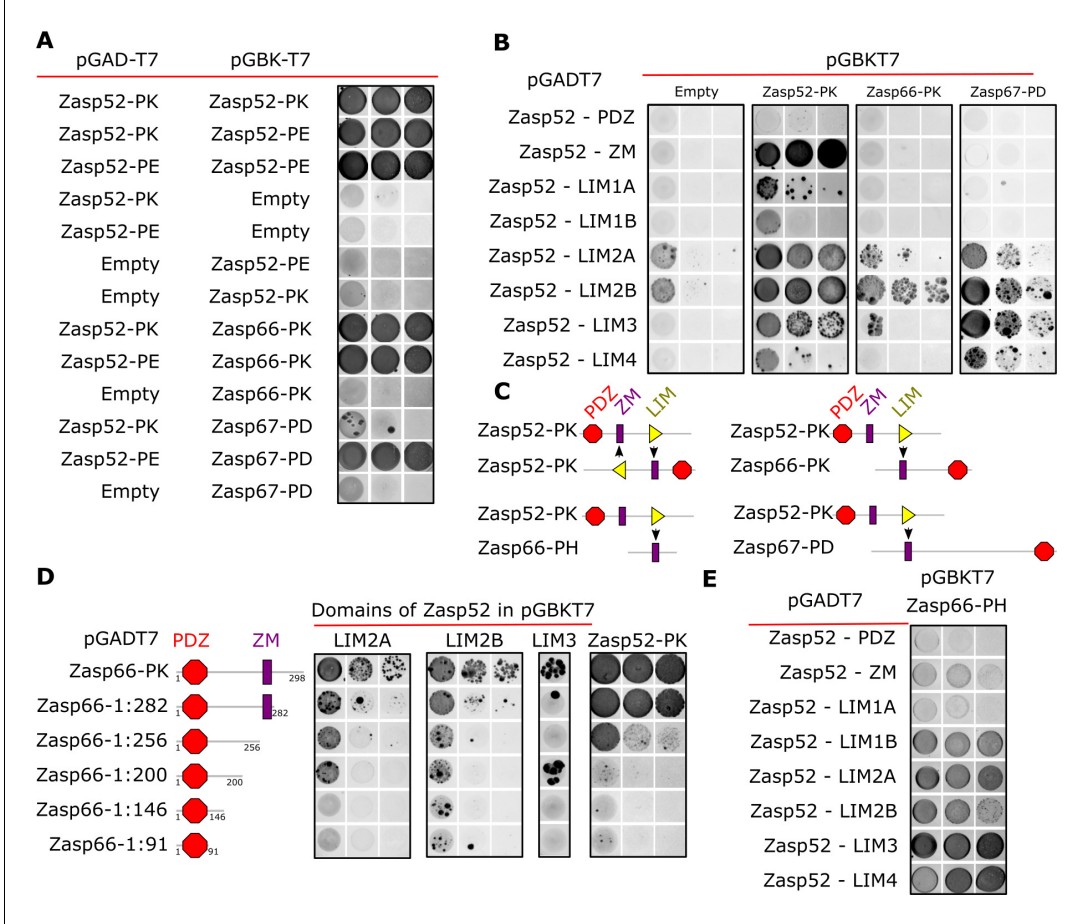

**Figure 2.** Y2H assays reveal domains involved in homo/heterodimerization. (**A**) Yeast two-hybrid assays between the three Zasp paralogs. Images of double-transformed yeast grown in selective -Ade/-His/-Leu/-Trp plates. Serial dilutions are shown from left to right (OD: 0.1, OD: 0.01, and OD: 0.001). Zasp52-PK dimerizes with itself and with the longer isoform Zasp52-PE. Zasp66 interacts with Zasp52-PK and Zasp52-PE. Zasp67 interacts with Zasp52-PK and Zasp52-PE. Negative controls using the DNA-binding domain (bait, pGAD-T7) or the Activating domain of Gal4 (prey, pGBK-T7) are shown. (**B**) Y2H assays testing the interaction between all protein domains encoded by the Zasp52 gene and Zasp52-PK, Zasp66-PK, or Zasp67-PD proteins. Zasp52-PK interacts with isolated ZM and LIM domains, Zasp66-PK and Zasp67-PD interact only with some LIM domains. LIM1A, LIM1B, LIM2A, and LIM2B are different splice isoforms of LIM1 and LIM2 domains. (**C**) Proposed model of Zasp homo/heterodimerization. The LIM domains bind the ZM domain. Zasp52-PK dimerization occurs through two ZM/LIM pairs. The heterodimerization between Zasp52-PK and Zasp66-PK or Zasp67-PD occurs through only one ZM/LIM-binding site. Zasp66-PH is a small isoform that only contains a ZM domain. (**D**) Y2H assays mapping the interaction between different Zasp52 LIM domains to Zasp66-PK using truncation mutants. (**E**) Y2H assays testing the interaction between the ZM only isoform Zasp66-PH, and the individual protein domains encoded by Zasp52.

The online version of this article includes the following figure supplement(s) for figure 2:

**Figure supplement 1.** Zasp self-interaction by co-IP and chemical crosslinking.

through its LIM domains, consistent with the Y2H results. α-Actinin localization at the Z-disc was not decreased in any of the Zasp mutants (*Figure 3—figure supplement 1B,C*).

To determine if LIM and ZM domains directly interact in flies at the Z-disc, we employed bimolecular fluorescence complementation (BiFC) assays (*Figure 3—figure supplement 1D*) (*Ciruela, 2008*; *Gohl et al., 2010*). We first confirmed that the ZM-only isoform Zasp66-PH fused to a Venus tag localized to Z-discs (*Figure 3—figure supplement 1E*). We then generated transgenic flies expressing Zasp52-PK and Zasp66-PH tagged with either the C-terminus or the N-terminus of yellow fluorescent protein (NYFP or CYFP) and quantified the fluorescence of reconstituted YFP at the Z-disc. Zasp52-PK-NYFP and Zasp52-PK-CYFP show specific fluorescence at the Z-disc, whereas controls do not (*Figure 3E–G*). More importantly, the ZM-only isoform Zasp66-PH-NYFP interacts specifically

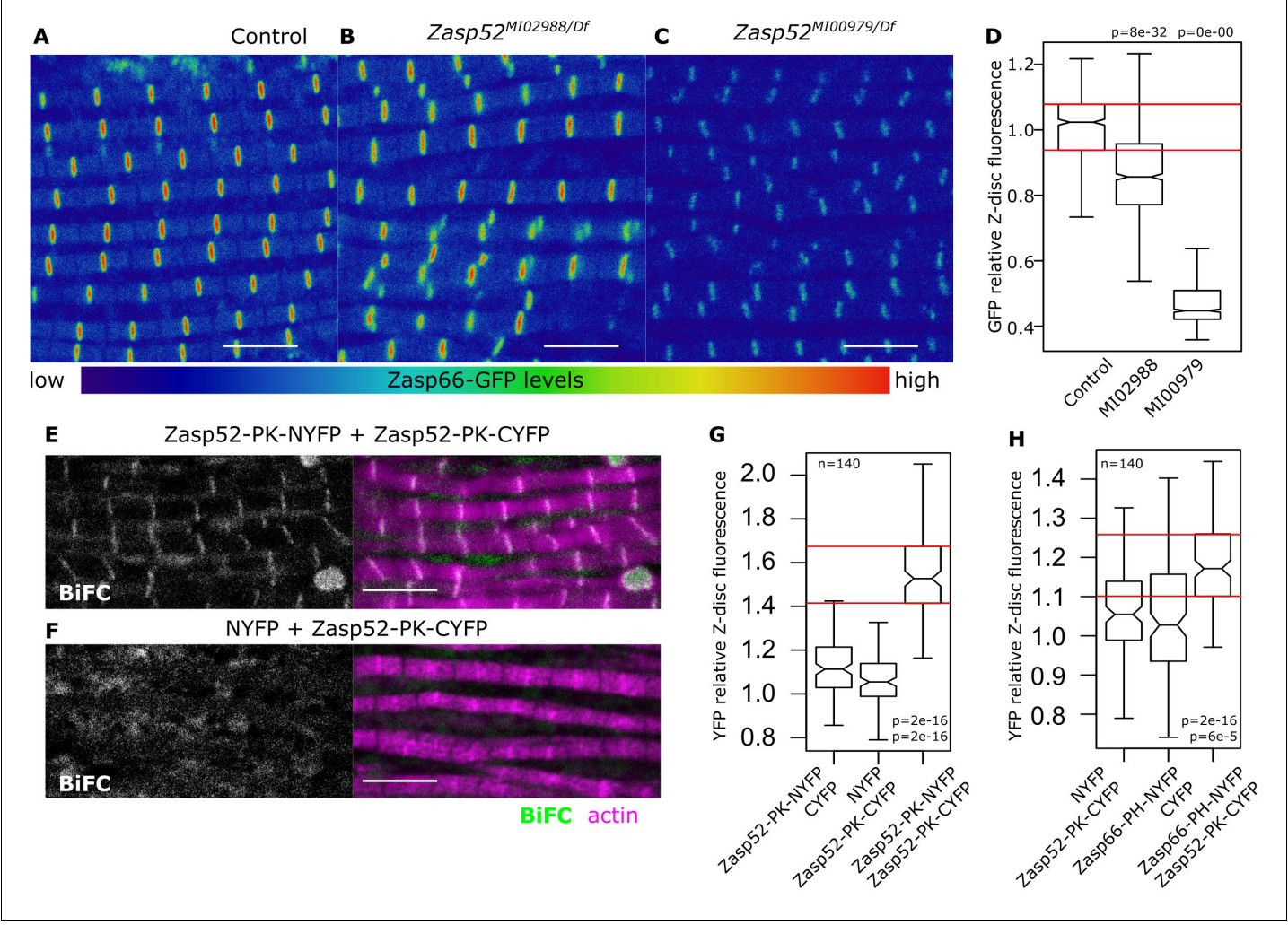

**Figure 3.** Zasp interaction in vivo at the Z-disc. (A–C) Confocal images of Zasp66-GFP IFM in control and *Zasp52* mutant backgrounds. Zasp66-GFP levels are lower in *Zasp52^{MI00979}* mutant than in the control or in *Zasp52^{MI02988}* mutants. (D) Boxplot of Zasp66-GFP intensities in control and *Zasp52* mutant backgrounds. (E, F) Examples of a negative BiFC control (F) and a positive BiFC signal (E) suggesting Zasp52-PK dimerizes at the Z-disc. (G, H) Plots of the BiFC fluorescence intensity values relative to background noise. Positive BiFC fluorescence is detected between Zasp52-PK and Zasp52-PK (G) and between Zasp52-PK and Zasp66-PH (H). Act88F-Gal4 was used to drive expression of NYFP- or CYFP-tagged proteins. Scale bar, 5 µm. p-Values in panels D, G, and H were calculated using Welch's two-sample t-test.

The online version of this article includes the following source data and figure supplement(s) for figure 3:

**Source data 1.** Zasp66 levels and BiFC values.

**Figure supplement 1.** *Zasp52* gene map and BiFC.

with Zasp52-PK-CYFP at the Z-disc (*Figure 3H*). Thus, ZM-LIM interaction of Zasp proteins occurs in vivo at the Z-disc.

## ZM domains and canonical Z-discs co-appeared in bilateral animals

LIM domains are well-known protein-protein interaction domains (*Kadrmas and Beckerle, 2004*). In contrast, functional information on the ZM domain is scarce. It is an uncharacterized short motif of 26 amino acids found only in Alp/Enigma proteins (*Klaavuniemi et al., 2004*; *Letunic et al., 2015*). Alignment of *Drosophila* Zasp-encoding genes to the ZM consensus sequence shows weak conservation (*Figure 4A*). To determine the phylogenetic distribution of the ZM domain, we compiled the presence of ZM domains in all branches of metazoans using data from the PFAM database (*Bateman et al., 2004*; *El-Gebali et al., 2019*). In contrast to the universal presence of LIM domains in eukaryotes, the ZM domain is only present in a subset of animal genomes and absent from plants,

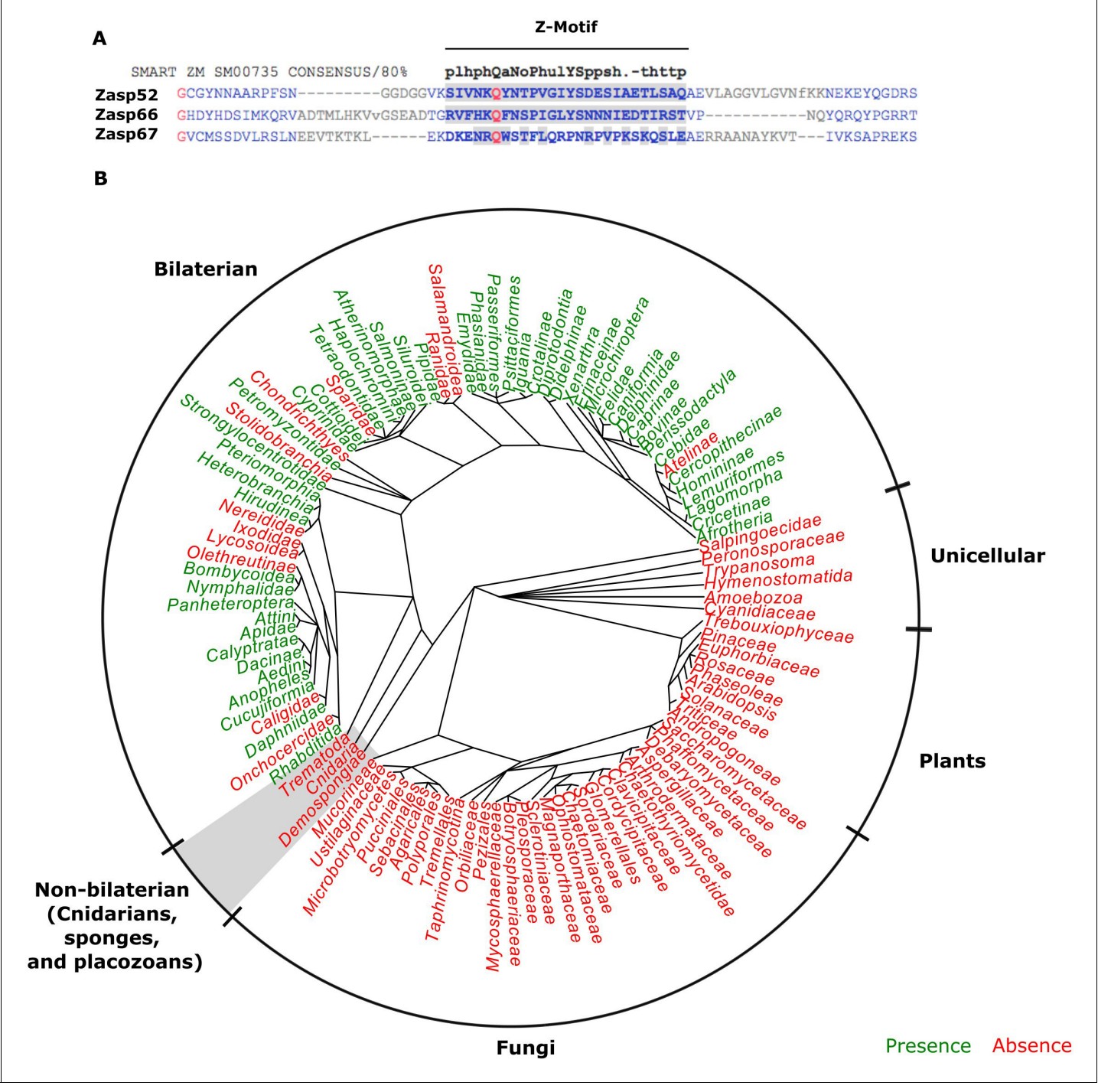

**Figure 4.** The evolution of the ZM domain. (A) Protein sequence alignment between the three Zasp paralogs in flies and the ZM consensus sequence SMART:SM00735. Similar amino acids to the consensus are highlighted in gray. (B) Radial phylogenetic tree of selected eukaryotic lineages with annotated presence of the ZM domain from the PFAM domain database (PFAM: PF15936). The ZM domain is restricted to bilateral animals.

fungi and unicellular metazoans (*Figure 4B*). Muscle striation evolved independently in cnidarians and bilaterians, with only the latter showing a canonical Z-disc structure (*Steinmetz et al., 2012*). Intriguingly, the ZM domain is restricted to bilateral animals (*Figure 4B*). Thus, our correlative data suggest that canonical Z-discs and ZM domains arose together during evolution.

## Differential temporal expression and localization of Zasp growing and blocking isoforms

Zasp isoforms can be divided into isoforms with multiple LIM domains (e.g. Zasp52-PR) and isoforms with just one or no LIM domain (e.g. Zasp52-PK, Zasp52-PP, Zasp66, Zasp67). We named the former growing isoforms and the latter blocking isoforms. We then considered what mechanism might coordinate Z-disc growth: we hypothesize that multiple LIM domain-containing Zasp proteins recruit other Zasp proteins by interacting with their ZM domains. If the recruited proteins are also multivalent (contain two or more LIM domains), more Zasp proteins will be recruited leading to Z-disc growth. However, if recruited proteins lack LIM domains, they block further recruitment of Zasp proteins, and Z-disc growth terminates (*Figure 5A*).

To regulate Z-disc growth, growing isoforms should be overrepresented at earlier stages of Z-disc formation, and blocking isoforms at later stages, to stop Z-disc growth. We used an RNAseq dataset from developing IFM that covers the whole Z-disc formation process, from 16 hr after puparium formation (APF) to newly eclosed flies (*Spletter et al., 2018*). We observed earlier expression of the growing isoforms compared to the blocking isoforms (*Figure 5B*). For example, multivalent Zasp52 growing isoforms are already strongly expressed at 24 hr APF, whereas some blocking isoforms without any LIM domains are expressed only after 60 hr APF (*Figure 5B*).

Z-discs grow at the periphery of the disc, starting as a small Z-body (*Orfanos et al., 2015*; *Shwartz et al., 2016*). Our model therefore predicts growing isoforms to be enriched at the centre and blocking isoforms to be enriched at the periphery of the final-sized Z-disc. To analyze the distribution of Zasp proteins within Z-discs, we made cross sections of myofibrils and evaluated the localization pattern of blocking isoforms (using Zasp66-GFP and Zasp67-GFP), and growing isoforms (using Zasp52-GFP; *Zasp52ZCL423*) at the level of the Z-disc. To label the entire disc, we used actin staining as counterstain (*Figure 5C*). To compensate for the low signal-to-noise ratio, we used a smoothening algorithm that takes advantage of the geometrical properties of discs (*Figure 5—figure supplement 1*). Actin is evenly distributed in the Z-disc (*Figure 5D,E*). The growing isoforms are present throughout the disc, but most concentrated at the centre of the Z-disc (*Figure 5D,E*). The blocking isoforms Zasp66 and Zasp67 are partially excluded from the centre of the disc and form a ring-like pattern (*Figure 5D,E*). Zasp66 localizes more at the periphery compared to Zasp67, which correlates with its later expression peak compared to Zasp67 (middle panels of *Figure 5B*).

## The balance between growing and blocking isoforms controls Z-disc growth and aggregation

We then asked whether decreasing the levels of growing isoforms would result in smaller Z-discs. We measured individual Z-discs in control and *Zasp52* mutants (*Figure 6A–C*). Interestingly, only the *Zasp52MI00979* mutant, which deletes multiple LIM domains, had smaller Z-discs compared to the control (*Figure 6B,C*). In *Zasp52MI00979*, the smaller size categories were significantly elevated, whereas *Zasp52MI02988* was comparable to the control (*Figure 6C*).

We next determined if the opposite approach, increasing the levels of blocking isoforms, also results in smaller Z-discs. Overexpressing the Zasp52-PP blocking isoform led to smaller Z-discs than the control (*Figure 6D*). This effect requires the ZM domain, because overexpression of Zasp52-Stop143 containing only a functional extended PDZ domain, causes a milder phenotype than Zasp52-PP (*Figure 6D*, *Figure 1—figure supplement 1*). Like Zasp52-PP, overexpression of Zasp66 or Zasp67 resulted in smaller Z-discs (*Figure 6E*). We also confirmed these phenotypes by measuring the relative fluorescence of Zasp52-mCherry at the Z-disc. Upon overexpression of Zasp52-PP, but not of Zasp52-Stop143, Zasp52-mCherry recruitment to the Z-disc is reduced (*Figure 6F*, *Figure 6—figure supplement 1A*). Likewise, overexpression of Zasp66 and Zasp67 reduces Zasp52-mCherry recruitment to the Z-disc (*Figure 6G*, *Figure 6—figure supplement 1A*). In contrast, α-actinin levels were unchanged in Zasp overexpression conditions (*Figure 6—figure supplement 1B,C*). These data indicate that increasing the ZM/LIM ratio in IFM results in smaller Z-discs.

Finally, we wondered if depleting the blocking isoforms encoded by *Zasp66* and *Zasp67* can generate bigger Z-discs or aggregates using CRISPR *Zasp66* and *Zasp67* null mutants (*González-Morales et al., 2019*). We observed rare aggregates in *Zasp66* and *Zasp67* single mutants (*Figure 7A,B,D*). As these two proteins likely have some redundant roles, we also analyzed the *Zasp66 Zasp67* double mutant, where we observed frequent aggregates (*Figure 7C,D*). Additionally,

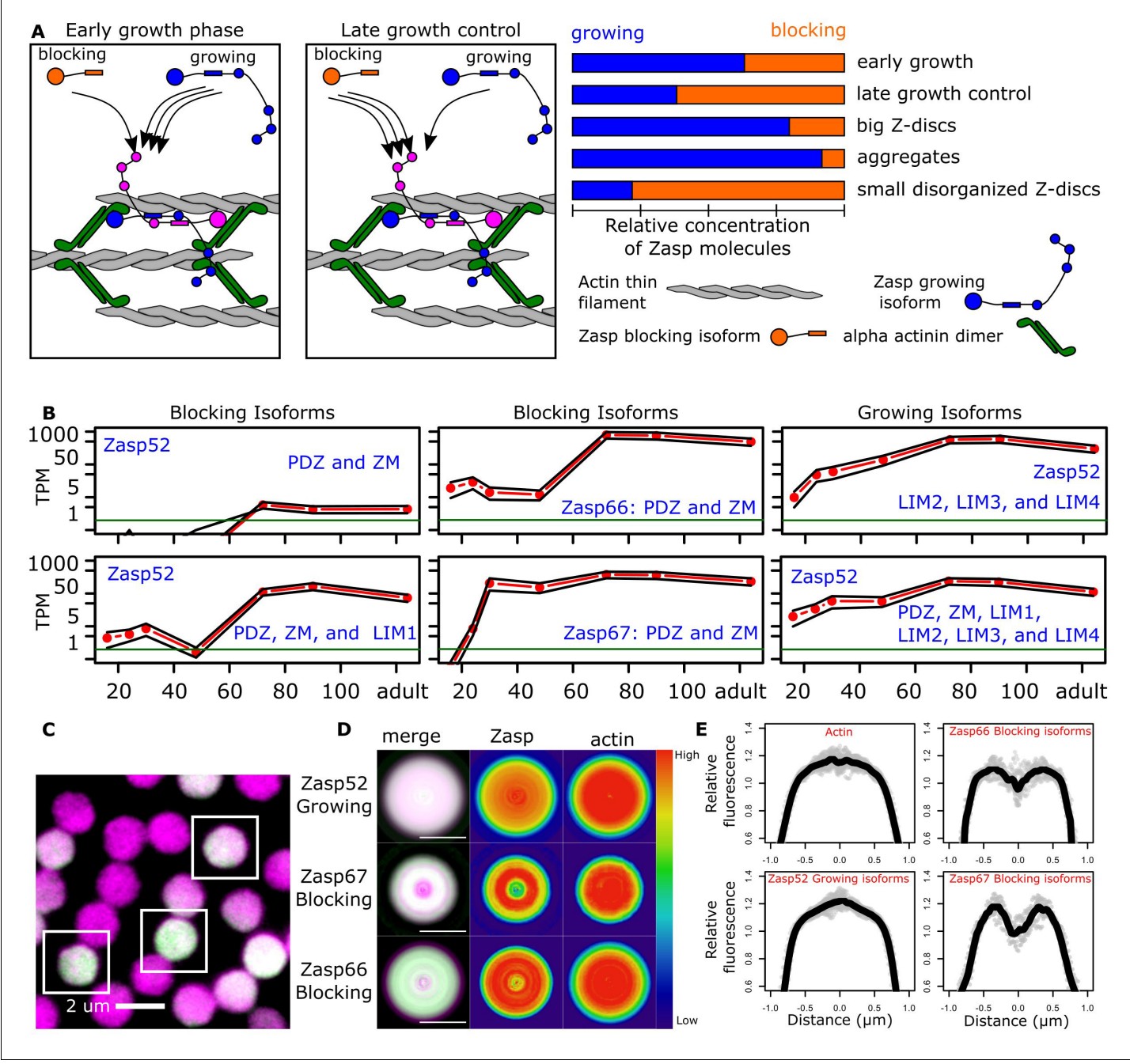

**Figure 5.** Balance of ZM to LIM domains sets the diameter of the Z-disc. (**A**) Proposed model of Z-disc growth and aggregate formation. In nascent Z-bodies, Zasp binds α-actinin and forms a homodimer between its ZM and LIM domains. During the early growth phase, Zasp growing isoforms use their free LIM domains to recruit more Zasp molecules through the ZM domain, thus growing the Z-disc. At later stages, growth is downregulated by the incorporation of blocking isoforms of Zasp, which are recruited to the Z-disc but cannot recruit further proteins because they lack LIM domains. (**B**) RNAseq data plots showing the expression of selected Zasp isoforms at different developmental timepoints. The y-axis corresponds to the log of the number of transcripts per million (TPM) and the x-axis to hours after pupa formation. Isoforms were classified as blocking isoforms (0 or 1 LIM domain) or growing isoforms (two or more LIM domains). (**C**) Confocal microscopy image of IFM cross sections expressing Zasp52-GFP (green) and stained for actin (magenta). (**D**) Differential distribution of Zasp growing and blocking isoforms throughout the Z-disc. Images of false-colored denoised Z-discs. Zasp52-growing isoforms are more concentrated at the center of the disc, Zasp66 and Zasp67 are mostly concentrated in the periphery. Color scale is shown at the right. (**E**) Profile plots of relative intensity values at the Z-disc diameters from at least 10 individual Z-discs. Growing isoforms peak at the center of the Z-discs, blocking isoforms have two peaks. Scale bars in D, 1 μm.

The online version of this article includes the following source data and figure supplement(s) for figure 5:

**Source data 1.** Line intensity profile plots.

*Figure 5 continued on next page*

*Figure 5 continued*

**Figure supplement 1.** Rotation smoothing of cross section Z-disc images.

the single mutants had normal-sized Z-discs, whereas the double mutant had enlarged Z-discs (*Figure 7E*). Thus, blocking isoforms are required to prevent Z-disc overgrowth.

As many diseases are believed to be caused by the formation of aggregates (*Baba et al., 1998*; *Selcen, 2008*), we asked if we can suppress aggregate formation in our model of overexpressed

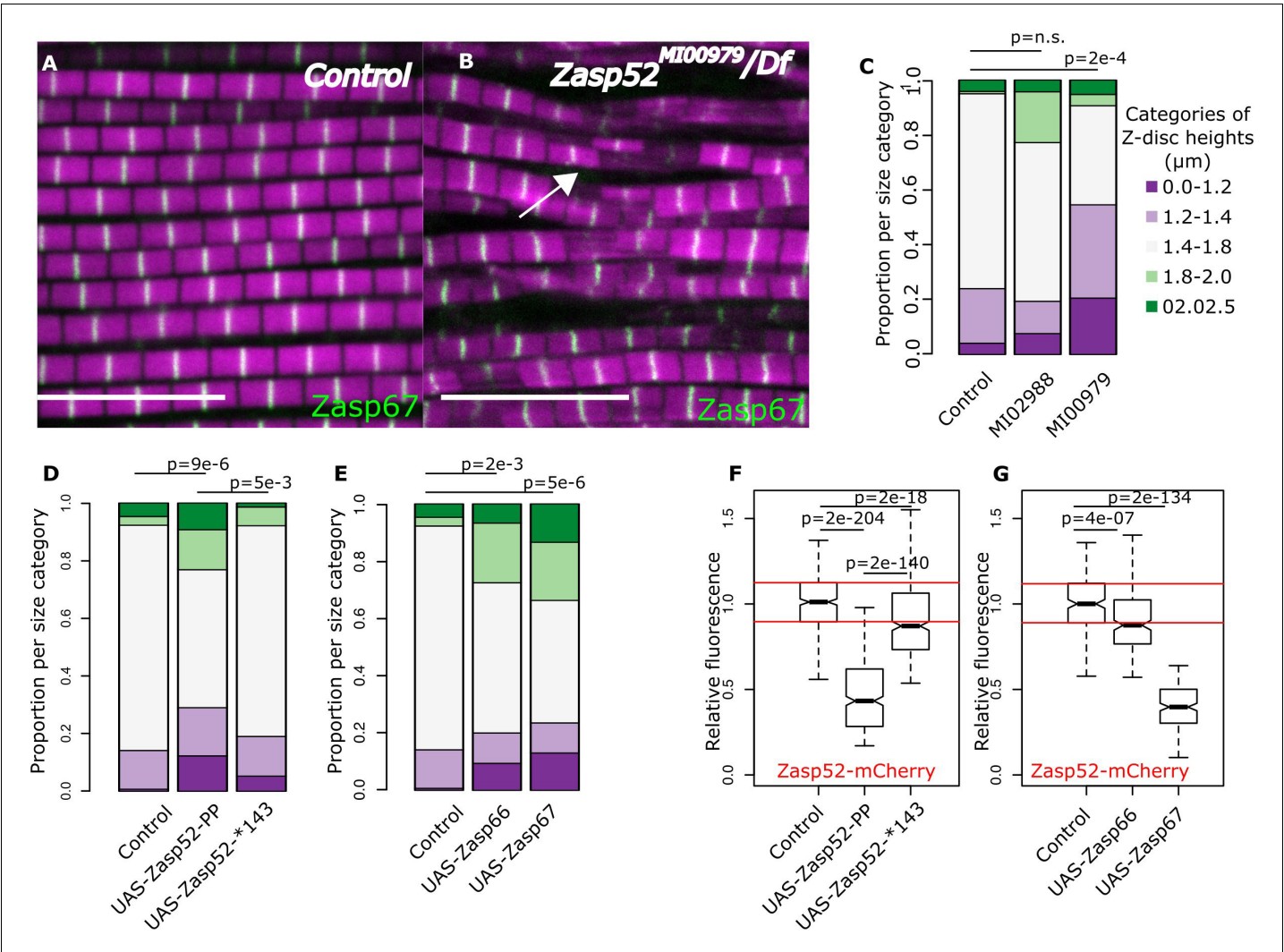

**Figure 6.** Balance of isoforms dictates Z-disc size. (**A and B**) Confocal microscopy images of control and *Zasp52^MI00979^* mutant. Actin filaments are marked in magenta and Z-discs in green. The *Zasp52^MI00979^* mutant has smaller and frayed Z-discs. (**C**) Frequency plot of Z-disc sizes in control, *Zasp52^MI02988^*, and *Zasp52^MI00979^* mutants. Small Z-discs are only observed in the *Zasp52^MI00979^* mutant. (**D**) Overexpression of the Zasp52-PP blocking isoform also results in smaller Z-discs. The small Z-disc phenotype is not observed in Zasp52-Stop143, lacking the ZM domain. (**E**) Small Z-disc phenotypes are observed upon overexpression of Zasp66 and Zasp67. (**F and G**) Boxplots of the Zasp52 fluorescence intensities upon overexpression of different blocking isoforms. (**F**) Zasp52-mCherry levels decrease upon overexpression of Zasp52-PP but are restored if the ZM domain of Zasp52-PP is deleted (Zasp52-*143). (**G**) Zasp52-mCherry levels also decrease upon overexpression of Zasp66 and Zasp67. Act88F-Gal4 was used for overexpression experiments in panels D-G. Scale bars, 5 μm. p-Values in panels C-E were calculated using Fisher's exact test for count data. p-Values in panels F and G were calculated using Welch's two-sample t-test.

The online version of this article includes the following source data and figure supplement(s) for figure 6:

**Source data 1.** Z-disc diameter estimates and Zasp52 levels.
**Figure supplement 1.** Actinin levels in different Zasp overexpression backgrounds.

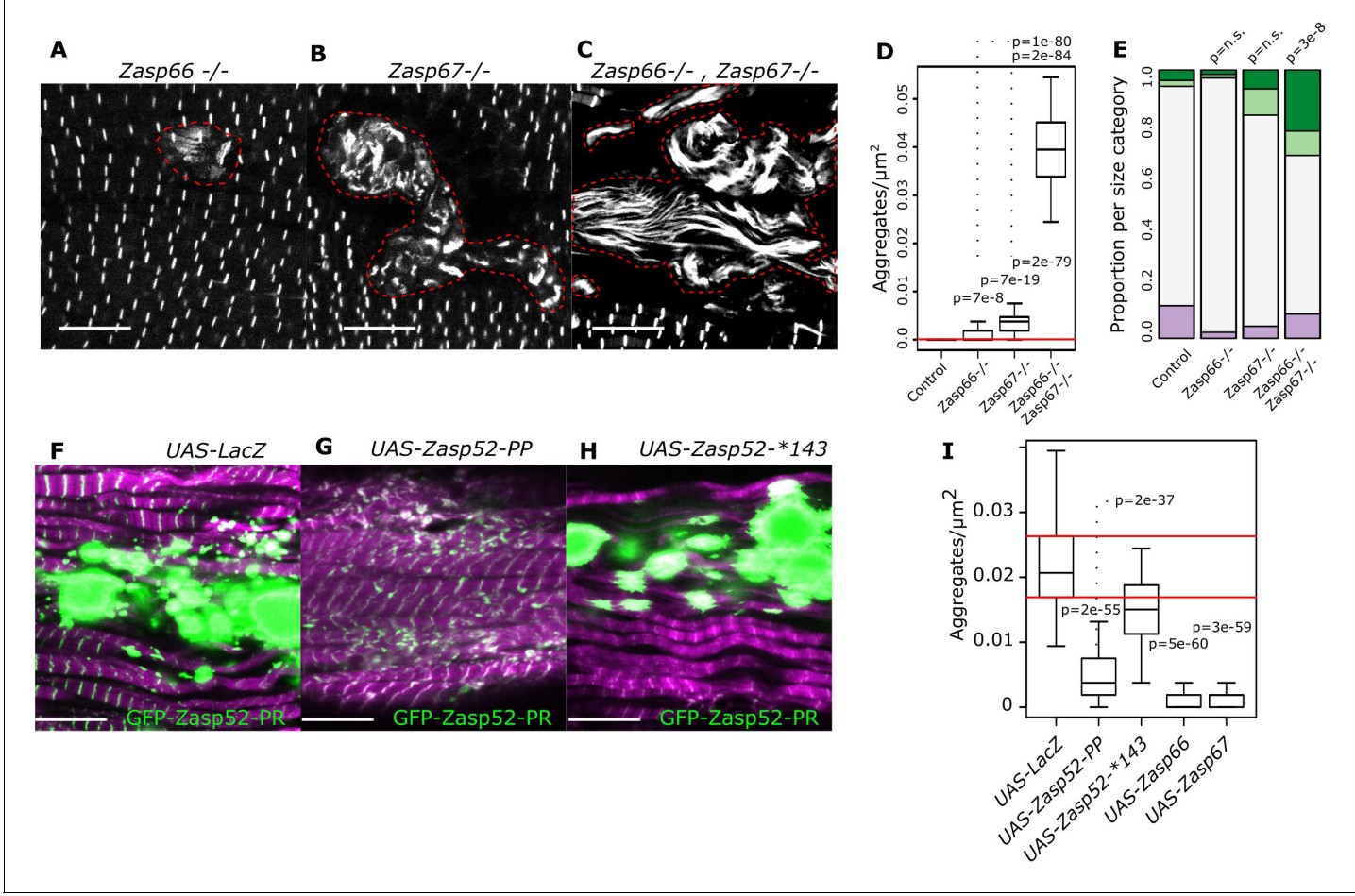

**Figure 7.** Isoform imbalance leads to Z-disc aggregation. (A–C) Confocal microscopy images of *Zasp66* and *Zasp67* single null mutants and the *Zasp66 Zasp67* double mutant. Aggregates are outlined by a stippled red line. (D) Plot of aggregate frequencies of single and double mutants. (E) Frequency plot of Z-disc size categories in single and double mutants (for categories see *Figure 6*). (F–H) Confocal microscopy images of muscles overexpressing the growing GFP-Zasp52-PR isoform and selected Zasp blocking isoforms or a LacZ control. (F) In the control, GFP-Zasp52-PR overexpression produces aggregates. (G) The aggregation phenotype is suppressed by co-overexpression of the blocking isoform Zasp52-PP. (H) The aggregation phenotype is not rescued upon co-overexpression of Zasp52-Stop143, which lacks the ZM domain. (I) Aggregation frequency in different co-overexpression backgrounds. Act88F-Gal4 was used for overexpression experiments in panels F-I. Scale bars in A-C and F-J, 10 μm. p-Values in panel E were calculated using Fisher's exact test for count data. p-Values in panels D and I were calculated using Welch's two-sample t-test.

The online version of this article includes the following source data for figure 7:

**Source data 1.** Z-disc diameter estimates and aggregate estimates.

GFP-Zasp52-PR (*Figure 1B*). Co-overexpression of both growing and the Zasp52-PP blocking isoform suppressed aggregate formation compared to a LacZ control (*Figure 7F–I*). The mutated blocking isoform Zasp52-Stop143 without the ZM domain did not suppress aggregation (*Figure 7H, I*). Co-overexpression of Zasp66 or Zasp67 also suppressed aggregation (*Figure 7I*).

## Discussion

How myofibrils assemble and grow until they reach their final, highly invariant size has remained poorly understood. Here, we reveal a conserved mechanism by which the Z-disc scaffold protein Zasp oligomerizes to induce Z-disc and thereby myofibril growth. Furthermore, we show that Z-disc

and sarcomere growth is terminated when shorter isoforms of Zasp are expressed that block multi-valency-driven oligomerization.

## Z-disc formation and growth is driven by multivalent oligomerization of Zasp proteins

Our findings indicate that Z-disc formation and growth is driven by multivalent oligomerization of Zasp proteins with multiple LIM domains, and eventually terminated at the proper Z-disc size by the upregulation of blocking isoforms without LIM domains (see model in *Figure 5A*). The process of Z body and Z-disc formation is reminiscent of membraneless organelles with compositions distinct from the surrounding cytosol, which form through a mechanism of phase separation (e.g. Cajal bodies or P bodies) (*Boeynaems et al., 2018*; *Weber, 2017*). Both have sharp boundaries between themselves and the cytoplasm, they form and organize as discrete puncta in the cytosol, and multi-valent protein domains are often involved in their formation (*Li et al., 2012*; *Weber, 2017*).

Sarcomere size is stereotyped in a given muscle type but distinct among different muscles (*Schönbauer et al., 2011*). How can our model explain differences in sarcomere sizes? The sarco-mere grows, while the Zasp growing isoforms dominate the Zasp isoform pool. Different sarcomere sizes can be achieved in two ways. First, the sarcomere growth period – the window of time in which Zasp growing isoforms dominate, might vary among muscle types. Second, the speed at which new Zasp molecules are recruited to the Z-disc, might be different among muscle types, while the growth period remains constant. Given the diversity of muscle types and therefore sarcomere sizes that exist, it is likely that a combination of these two strategies occur simultaneously.

Finally, apart from the ZM-LIM mechanism described here, additional redundant mechanisms to control Z-disc growth might exist, as evidenced by the observation that Zasp52-PR overexpression makes big Z-discs and aggregates, while the mutant removing Zasp52 LIM domains reduces Z-disc size to a comparatively small degree. Redundant mechanisms might operate through other LIM domain proteins, or the coordination of Z-disc and M-line growth, all of which may provide important buffering functions to ensure proper myofibril size, which is crucial for fully functional muscles.

## The role of the ZM domain

The ZM domain is a conserved domain without a clearly defined function. On its own, the ZM domain from two mouse Zasp proteins localizes to the Z-disc (*Klaavuniemi et al., 2004*; *Klaavuniemi and Ylänne, 2006*). Our data suggest that Z-disc localization is a conserved feature of ZM domains from vertebrates to insects. ZM-containing proteins are tethered to the Z-disc by the physical interaction with the LIM domains of other Zasp proteins. In sum, the LIM domain serves as a recruitment signal for Zasp proteins and potentially other unidentified ZM-containing proteins to join the Z-disc. In addition, given the appearance of the ZM domain in bilateral animals with canonical Z-discs, we postulate that a conserved mechanism involving LIM-ZM binding underlies Z-disc growth and growth termination.

## Blocking and growing isoforms in vertebrates

In vertebrates, the Zasp proteins are very diverse and are better known as Alp/Enigma family: ZASP/Cypher/Oracle/LDB3/PDLIM6, ENH/PDLIM5, PDLIM7/ENIGMA/LMP-1, CLP36/PDLIM1/Elfin/hCLIM1, PDLIM2/Mystique/SLIM, ALP/PDLIM3, and RIL/PDLIM4. The ZM/DUF4749 motif occurs in ZASP, CLP36, PDLIM2, ALP and RIL (*Cheng et al., 2010*; *D'Cruz et al., 2016*; *Faulkner et al., 1999*; *Vallenius et al., 2004*; *Zheng et al., 2010*; *Zhou et al., 2001*). The LIM domain occurs in all Zasp proteins, either as one domain in Alp family members or as three domains in Enigma family members (*Zheng et al., 2010*). We identified two Zasp genes that encode only blocking isoforms in fruit flies: *Zasp66* and *Zasp67*, and one gene, *Zasp52*, that encodes blocking and growing isoforms. Although *Zasp66* and *Zasp67* genes are insect-specific (*González-Morales et al., 2019*), vertebrate Alp/Enigma genes also express isoforms without LIM domains that could fulfill a blocking isoform function (*Cheng et al., 2011*; *Zheng et al., 2010*). In addition, because Zasp52-PK, which only contains one LIM domain, behaves as a blocking isoform, the Alp members with only one LIM domain might also behave as blocking isoforms.

The function of the growing isoforms of Zasp requires multiple functional LIM domains. As the Enigma family members contain three C-terminal LIM domains, they are the ideal candidates to fulfill

the growing role in vertebrates. Three Enigma proteins exist in vertebrates: PDLIM7/Enigma/LMP-1, ENH/PDLIM5 and ZASP/Cypher/Oracle/LDB3/PDLIM6. Functional redundancy between them at the Z-disc is likely common and demonstrated in one case (*Mu et al., 2015*). In Cypher knockout mice sarcomere assembly occurs normally during development, followed by immediate sarcomere failure after postnatal onset of contractility (*Zhou et al., 2001*). ENH mutants exhibit cardiac dilation and abnormal Z-disc structure in the heart (*Cheng et al., 2010*). Intriguingly, in both Cypher and ENH single mutants, as well as Cypher ENH double mutants, sarcomeres look considerably smaller in diameter in electron microscopy images (*Cheng et al., 2010*; *Mu et al., 2015*; *Zhou et al., 2001*). Thus, a similar role for Enigma proteins in setting sarcomere diameter in vertebrates appears likely.

### Sarcomere size control in human myopathies

Is the mechanism that controls Z-disc size related to the protein aggregation defects in human myopathies? Our Z-disc oligomerization hypothesis agrees well with the observation that many myopathies present aggregates, and several human ZASP mutations have been linked to aggregate-forming myopathies (*Murphy and Young, 2015*; *Selcen and Engel, 2005*). Many ZASP mutations linked to disease lie within the ZM domain or one of the LIM domains (*Selcen and Engel, 2005*; *Theis et al., 2006*; *Vatta et al., 2003*). Protein aggregation in myopathy patients might be a consequence of an imbalance in the mechanism that controls sarcomere size, favoring the growing over the blocking isoforms. If this were the case, our data points to a potential therapeutic avenue: blocking the growing isoforms with short peptides containing a ZM domain.

In conclusion, we propose that a conserved mechanism involving LIM-ZM binding underlies Z-disc growth and therefore myofibril diameter.

## Materials and methods

**Key resources table**

| Reagent type (species) or resource | Designation | Source or reference | Identifiers | Additional information |
|---|---|---|---|---|
| Gene (*Drosophila melanogaster*) | Zasp52 | | FBgn0265991 | |
| Gene (*Drosophila melanogaster*) | Zasp66 | | FBgn0035917 | |
| Gene (*Drosophila melanogaster*) | Zasp67 | | FBgn0036044 | |
| Gene (*Drosophila melanogaster*) | Actn | | FBgn0000667 | |
| Genetic reagent (*Drosophila melanogaster*) | *Act88F-Gal4* | RM Cripps PMID: 22008792 | FBal0268407 | |
| Genetic reagent (*Drosophila melanogaster*) | *UH3-Gal4* (*P[GawB]HkUH3*) | Anja Katzemich PMID: 23505387 | FBti0148868 | |
| Genetic reagent (*Drosophila melanogaster*) | *UAS-LacZ* | BDSC | RRID:BDSC_3356 | |
| Genetic reagent (*Drosophila melanogaster*) | *UAS-GFP-Zasp52-PR* | Current study | N/A | Expresses full length Zasp52-PR isoform with a N terminal GFP tag under UAS. The landing site is ZH-attP-86Fb. |
| Genetic reagent (*Drosophila melanogaster*) | *UAS-GFP-Zasp52-PK* | Current study | N/A | Expresses Zasp52-PK isoform with a N terminal GFP tag under UAS. The landing site is ZH-attP-86Fb. |

*Continued on next page*

Continued

| Reagent type (species) or resource | Designation | Source or reference | Identifiers | Additional information |
|---|---|---|---|---|
| Genetic reagent (Drosophila melanogaster) | UAS-Flag-Zasp52-PR | Kuo An Liao PMID: 27783625 | FBal0323349 | Expresses full length Zasp52-PR isoform with a N terminal Flag tag under UAS. The landing site is ZH-attP-86Fb. |
| Genetic reagent (Drosophila melanogaster) | UAS-Flag-Zasp52-PR-ΔPDZ | Kuo An Liao PMID: 27783625 | FBal0323350 | Expresses Full length Zasp52 without the PDZ domain and a N terminal Flag tag under UAS. The landing site is ZH-attP-86Fb. |
| Genetic reagent (Drosophila melanogaster) | UAS-Flag-Zasp52-PR-ΔZM | Kuo An Liao PMID: 27783625 | FBal0323351 | Expresses full length Zasp52-PR isoform without the ZM domain and a N terminal Flag tag under UAS. The landing site is ZH-attP-86Fb. |
| Genetic reagent (Drosophila melanogaster) | UAS-Flag-Zasp52-PP | Current study | N/A | Expresses smallest Zasp52 isoform with a N terminal Flag tag under UAS. The landing site is ZH-attP-86Fb. |
| Genetic reagent (Drosophila melanogaster) | UAS-Flag-Zasp52-Stop143 | Current study | N/A | Expresses Zasp52-PK isoform with a N terminal Flag tag and a stop codon at position 143 under UAS. The landing site is ZH-attP-86Fb. |
| Genetic reagent (Drosophila melanogaster) | UAS-Zasp66-PK-Flag-HA | Current study | N/A | Transgenic made from the UFO11045 plasmid from DGRC. The landing site is ZH-attP-58A. |
| Genetic reagent (Drosophila melanogaster) | UAS-Zasp67-PE-Flag-HA | Current study | N/A | Transgenic from Zasp67 sequence synthesized by Genscript and cloned into a pUASattb vector. The landing site is ZH-attP-58A. |
| Genetic reagent (Drosophila melanogaster) | Zasp52-MI02988-mCherry | Nicanor Gonzalez-Morales PMID: 29423427 | PMID: 29423427 | Replacement of the MIMIC02988 cassette in Zasp52 with an in-frame mCherry tag. |
| Genetic reagent (Drosophila melanogaster) | Zasp52-GFP Zasp52[ZCL423] | BDSC | RRID:BDSC_58790 | |
| Genetic reagent (Drosophila melanogaster) | Zasp66-GFP Zasp66[ZCL0663] | BDSC | RRID:BDSC_6824 | |
| Genetic reagent (Drosophila melanogaster) | Zasp67-GFP fTRG | VDRC | v318355 | |
| Genetic reagent (Drosophila melanogaster) | UAS-Actn-KK (RNAi) | VDRC | v110719; FBst0482284 | |
| Genetic reagent (Drosophila melanogaster) | Zasp52[MI02988] | BDSC | RRID:BDSC_41034 | |

*Continued*

| Reagent type (species) or resource | Designation | Source or reference | Identifiers | Additional information |
|---|---|---|---|---|
| Genetic reagent (*Drosophila melanogaster*) | *Zasp52[MI07547]* | BDSC | RRID:BDSC_43724 | |
| Genetic reagent (*Drosophila melanogaster*) | *Zasp52[MI00979]* | BDSC | RRID:BDSC_33099 | |
| Genetic reagent (*Drosophila melanogaster*) | *Zasp66[KO]* | PMID: 31123042 | PMID: 31123042 | CRISPR null mutant of Zasp66 |
| Genetic reagent (*Drosophila melanogaster*) | *Zasp67[KO]* | PMID: 31123042 | PMID: 31123042 | CRISPR null mutant of Zasp67 |
| Genetic reagent (*Drosophila melanogaster*) | *UAS-Zasp66-PH-Venus* | Current study | N/A | The Zasp66-PH isoform that contains only a ZM domain cloned into pBID-UAS-GV vector. The landing site is ZH-attP-58A. |
| Genetic reagent (*Drosophila melanogaster*) | *UAS-Zasp66-PH-NYFP (CYFP)* | Current study | N/A | The Zasp66-PH isoform fused to either NYFP or CYFP. |
| Genetic reagent (*Drosophila melanogaster*) | *UAS-Zasp52-PK-NYFP (CYFP)* | Current study | N/A | The Zasp52-PK isoform fused to either NYFP or CYFP. |
| Recombinant DNA reagent | Plasmid: pGADT7 | Clontech | 630442 | |
| Recombinant DNA reagent | Plasmid: pBD-Gal4-Zasp67 | DGRC | CT33772-BD | |
| Recombinant DNA reagent | Plasmid: pOAD-Zasp67 | DGRC | CT33772-AD | |
| Recombinant DNA reagent | Plasmid: pENTRY-Zasp52-PK | DGRC | DGRC: GEO02280 | |
| Recombinant DNA reagent | Plasmid: pENTRY-Zasp52-PE GEO12859 | DGRC | DGRC: GEO12859 | |
| Recombinant DNA reagent | Plasmid: pENTRY-Zasp66-PH GEO14752 | DGRC | DGRC: GEO14752 | |
| Recombinant DNA reagent | Plasmid: pGBKT7-GW | Addgene | 61703 | |
| Recombinant DNA reagent | Plasmid: pGADT7-GW | Addgene | 61702 | |
| Recombinant DNA reagent | Plasmid: pGADT7-Zasp66-PK | Current study | N/A | Zasp66-PK isoform cloned into pGADT7. |
| Recombinant DNA reagent | Plasmid: pGADT7-Zasp66-PK with stop codons | Current study | N/A | Stop codons were introduced in the Zasp66-PK-pGADT7 plasmid by site-directed mutagenesis (Genscript). |
| Recombinant DNA reagent | Plasmid: pGBKT7-Zasp52 individual domains: PDZ, ZM, LIM1a, LIM1b, LIM2a, LIM2b, LIM3 and LIM4 | Current study | N/A | All individual domains of Zasp52 cloned into pGBKT7. |

*Continued on next page*

*Continued*

| Reagent type (species) or resource | Designation | Source or reference | Identifiers | Additional information |
|---|---|---|---|---|
| Recombinant DNA reagent | Plasmid: pGBKT7 GW-Zasp52-PK | Current study | N/A | Zasp52-PK isoform cloned into pGBKT7GW using Gateway cloning. |
| Recombinant DNA reagent | Plasmid: pGADT7 GW-Zasp52-PK | Current study | N/A | Zasp52-PK isoform cloned into pGADT7GW using Gateway cloning. |
| Recombinant DNA reagent | Plasmid: pGBKT7 GW-Zasp52-PE | Current study | N/A | Zasp52-PE isoform cloned into pGBKT7GW using Gateway cloning |
| Recombinant DNA reagent | Plasmid: pGADT7 GW-Zasp52-PE | Current study | N/A | Zasp52-PE isoform cloned into pGADT7GW using Gateway cloning. |
| Recombinant DNA reagent | Plasmid: pGBKT7GW-Zasp66-PH | Current study | N/A | Zasp66-PH isoform cloned into pGBKT7GW using Gateway cloning |
| Recombinant DNA reagent | Plasmid: pGADT7GW-Zasp66-PH | Current study | N/A | Zasp66-PH isoform cloned into pGADT7GW using Gateway cloning. |
| Recombinant DNA reagent | Plasmid: pDEST-pUAS-RfB-HA-CYFP-attB | Sven Bogdan PMID: 20937809 | FBrf0212496 | |
| Recombinant DNA reagent | Plasmid: pDEST-pUAS-RfB-myc-NYFP-attB | Sven Bogdan PMID: 20937809 | FBrf0212496 | |
| Strain, strain background (*Saccharomyces cerevisiae*) | Matchmaker Y2HGold | Clontech | 630498 | |
| Strain, strain background (*E. coli*) | BL21 | NEB | C2530H | |
| Antibody | Anti-Flag | SIGMA | F3165 | 1:200 |
| Software, algorithm | R Project for Statistical Computing: base and ape packages | https://cran.r-project.org/ | RRID:SCR_001905 | |
| Software, algorithm | Salmon | https://combine-lab.github.io/salmon/ | RRID:SCR_017036 | |
| Software, algorithm | ImageJ/Fiji distribution | https://fiji.sc/ | RRID: SCR_002285 | |
| Software, algorithm | Galaxy | https://usegalaxy.org/ | RRID:SCR_006281 | |
| Chemical compound, drug | ethylene glycol bis-sulfosuccinimidyl succinate (EGS) | Fisher Scientific | 21565 | |
| Chemical compound, drug | YPDA medium | Clontech | 630464 | |
| Chemical compound, drug | Minimal SD Base | Clontech | 630411 | |
| Chemical compound, drug | -Leu /- Trp DO Supplement | Clontech | 630417 | |
| Chemical compound, drug | -Ade /- His /- Leu/-Trp DO Supplement | Clontech | 630428 | |
| Chemical compound, drug | Acti-stain 488 phalloidin | CYTOSKELETON, INC | PHDG1-A | |

*Continued on next page*

*Continued*

| Reagent type (species) or resource | Designation | Source or reference | Identifiers | Additional information |
|---|---|---|---|---|
| Chemical compound, drug | Alexa633-Phalloidin | Fisher Scientific | A22284 | |
| Chemical compound, drug | Rhodamine-phalloidin | Fisher Scientific | 10063052 | |

## Experimental model

We used *Drosophila melanogaster* as a model organism for most experiments. Fly stocks and crosses were raised at 25°C. A comprehensive list of all strains used and generated can be found in the Key Resources Table. We used *Saccharomyces cerevisiae* for the yeast two-hybrid assays. *Escherichia coli* BL-21 strain was used to express recombinant proteins.

Transgenic flies were generated by site-directed integration using the PhiC31 integrase method into either *M[3xP3-RFP.attP]ZH-58A-* or *M[3xP3-RFP.attP]ZH-86Fb*-bearing flies to ensure comparable expression levels (*Bischof et al., 2007*; *Markstein et al., 2008*). Unless stated otherwise, Act88F-Gal4 was used to drive strong transgene expression in the IFM (*Bryantsev et al., 2012*).

## Yeast two hybrid and protein crosslinking assays

Partial or complete coding sequences for Zasp52, Zasp66 and Zasp67 were cloned into Y2H vectors either by PCR-ligase cloning or through Gateway cloning. Constructs and cloning details are listed in the Key Resources Table. All constructs were verified by sequencing. Selected constructs were transformed into the Matchmaker Y2H Gold strain using the lithium acetate method (*Gietz and Schiestl, 2007*). Double transformant colonies were selected and amplified in media lacking leucine and tryptophan (-Leu/-Trp). Serial dilutions of the selected double transformants were grown in plates lacking leucine, tryptophan, histidine and adenine (-Ade/-His/- Leu/-Trp) to test for protein-protein interactions. The experiments were done at least three times.

Recombinant 6xHis-Zasp52-PK-FLAG was expressed in *Escherichia coli* BL21 bacteria. Then, 6xHis-Zasp52-PK-FLAG was purified from the protein extracts using Ni-NTA agarose beads (Qiagen) for 3 hr at 4°C. The purified protein was then dialyzed overnight at 4°C. Finally, purified 6xHis-Zasp52-PK-FLAG was either incubated with ethylene glycol bis-sulfosuccinimidyl succinate (EGS) or alone. Then, the protein samples were analyzed by denaturing SDS-PAGE followed by western blotting with anti-Flag antibody (1:5000).

## Muscle staining and microscopy

The IFM were dissected as previously described (*González-Morales et al., 2017*; *Xiao et al., 2017*). All images were acquired with comparable parameters. In agreement with previous studies, at least 10 thoraces were dissected for each condition. Samples were allocated into experimental groups according to their genotype. A big number of flies were collected for each experimental group and a subsample was randomly selected for dissection.

## Bimolecular fluorescence complementation assay

Zasp52-PK, Zasp52-PE and Zasp66-PH Gateway pENTRY clones were obtained from the DGRC (GEO02280, GEO12859 and GEO14752). The BIFC pDEST vectors pUAS-RfB-HA-CYFP-attB and pUAS-RfB-myc-NYFP-attB vectors were a kind gift from Sven Bogdan (*Gohl et al., 2010*). Constructs were cloned using Gateway technology, and the final vectors were verified by sequencing. The Key Resources Table contains a list of transgenic flies used in this study.

The Act88F-Gal4 driver line was used to express CYFP- and NYFP-tagged proteins in the IFM. Then, the YFP fluorescence of individual Z-discs was quantified using the ImageJ plot profile tool (*Schindelin et al., 2012*). At least 20 samples were used for each condition. The data was normalized to the basal noise levels and plotted in R software.

## Image analyses

To estimate the size of individual Z-discs and the fluorescence intensity levels of Z-disc proteins we first used a segmentation method that allows individual Z-discs to be measured (*Xiao et al., 2017*). Briefly, images from fluorescent Z-disc proteins were passed through a threshold filter, and then individual Z-discs were separated and measured using the analyze particles tool. Both tools can be found in any recent ImageJ/Fiji distribution (*Schindelin et al., 2012*). To estimate the Z-disc size, we measured the Z-disc height, that corresponds to the disc diameter. Then, the data was analyzed and plotted in R software. To estimate the relative fluorescence intensities of Z-disc proteins, we first obtained the mean intensity raw values for each Z-disc. Then, we normalized the values to a control genotype. The resulting data was then analyzed in R software. Rare segmentation mistakes result from Z-discs that are in close proximity. We could filter these mistakes out by size exclusion in the control images, but not in mutant conditions. We decided to leave the segmentation mistakes for consistency. If anything, this method underestimates the size differences between control and mutants.

Batch Macro processing code for measuring individual Z-disc side-views in ImageJ

```
setAutoThreshold("Default dark");
run("Analyze Particles...", "size = 0.2 Infinity show = Outlines display
exclude");
```

To estimate the density of aggregates, present in a given IFM sample, we used confocal images of GFP- or mCherry-tagged Zasp. Each 36 × 36 µm image was divided into 256 2 × 2 µm images. An automatic threshold filter based on the original image was then used on the small images, and the area above the given threshold was measured. Then, we used R software to count the number of images with aggregates.

Batch Macro processing code for aggregation estimates in ImageJ

```
run("Montage  to  Stack...",  "images_per_row  =  16  images_per_column  =  16
border = 0");
 macro "Measure Stack" {
saveSettings;
setOption("Stack position", true);
for (n = 1; n <= nSlices; n++) {
setSlice(n);
setAutoThreshold("Default dark stack");
//run("Threshold...");
run("Create Selection");;
run("Measure");
}
restoreSettings;
}
```

## Cross-section images of Z-discs

Thoraces of *Zasp52-GFP (Zasp52^{ZCL423}), Zasp66-GFP (Zasp66^{ZCL0663}),* and *Zasp67-GFP (Zasp67 ^{fTRG})* flies were fixed in paraformaldehyde and embedded into low-melting agarose blocks. Then, we sectioned the thoraces into 100-µm-thick sections using a standard vibratome. The sections were stained with TRITC-phalloidin and imaged using a LEICA SP8 confocal microscope and a 63x/1.4 oil objective. Images were then processed to isolate individual properly oriented Z-discs. To avoid artifacts from misoriented Z-discs, we selected only those in which the GFP signal coming from the Z-disc would cover the whole sarcomere area stained with phalloidin. To remove the background noise from cross-section Z-disc images, we took advantage of the geometrical properties of the Z-disc. We drew a selection circle around individual Z-discs and rotated the image by 2 degrees 180 times. Then, we calculated the average from all the rotated images.

We compared the Z-disc size categorical data using Fisher's exact test for count data, p-values above 0.001 were noted as not significant. We used Welch's two-sample t-test to compare the relative fluorescence intensities between groups. We used Welch's two-sample t-test to compare the aggregation frequency estimates.

## Phylogenetic distribution of ZM domain proteins

All species that contain at least one protein with either the ZM (PF15936) or the LIM (PF00412) domain were downloaded from the PFAM web (*Bateman et al., 2004*; *El-Gebali et al., 2019*). A general phylogenetic tree was plotted based on the NCBI taxonomy data (*Federhen, 2012*) using the *ape:plot.phyllo* tool (*Paradis, 2012*).

## RNAseq analysis

We uploaded all individual RNA-seq SRA reads from the project: PRJNA419412 (GEO: GSE107247) (*Spletter et al., 2018*). Each SRA read corresponds to a different time point in IFM development. We extracted the single-end reads using the fastq-dump tool. We then used Salmon v0.8.2 (*Patro et al., 2017*) to quantify the expression of all transcripts as Relative Transcripts Per Million (TPM) values. We used all the cDNAs from the *D. melanogaster* r6 reference transcriptome in fasta format (*Adams et al., 2000*). Then, the individual transcripts corresponding to the three Zasp genes were grouped according to their encoded protein domain architectures. Finally, we calculated the mean TPM values between similar isoforms and plotted the results over developmental time using R software.

The accession number for all reads used are: SRR1665023, SRR1665024, SRR1665025, SRR1665026, SRR1665027, SRR1665028, SRR6314253, SRR6314254, SRR6314255, SRR6314256, SRR6314257, SRR6314258, SRR6314259, SRR6314260, SRR6314261, SRR6314262, SRR6314263, SRR6314273, SRR6314274, SRR6314275, SRR6314276, SRR6314277, SRR6314278.

## Acknowledgements

We appreciate the help of Tuana Correia-Mesquita and Beili Hu in making transgenic flies. We thank the Bloomington Drosophila stock center, Vienna Drosophila Resource Center, and the Drosophila Genomics Resource Center for materials and the CIAN imaging facility for providing access to confocal microscopy. This work was supported by operating grants MOP-142475 and PJT-155995 from the Canadian Institutes of Health Research and by RGPIN 2016–06793 from the Natural Sciences and Engineering Research Council of Canada.

## Additional information

### Funding

| Funder | Grant reference number | Author |
| --- | --- | --- |
| Canadian Institutes of Health Research | MOP-142475 | Nicanor González-Morales<br>Yu Shu Xiao<br>Matthew Aaron Schilling<br>Océane Marescal<br>Kuo An Liao |
| Canadian Institutes of Health Research | PJT-155995 | Nicanor González-Morales<br>Yu Shu Xiao<br>Matthew Aaron Schilling<br>Océane Marescal<br>Kuo An Liao |
| Natural Sciences and Engineering Research Council of Canada | RGPIN 2016-06793 | Nicanor González-Morales<br>Yu Shu Xiao<br>Matthew Aaron Schilling<br>Océane Marescal<br>Kuo An Liao |

The funders had no role in study design, data collection and interpretation, or the decision to submit the work for publication.

## Author contributions
Nicanor González-Morales, Conceptualization, Data curation, Software, Formal analysis, Supervision, Validation, Investigation, Visualization, Methodology, Writing—original draft, Writing—review and editing; Yu Shu Xiao, Matthew Aaron Schilling, Océane Marescal, Kuo An Liao, Investigation, Methodology; Frieder Schöck, Conceptualization, Supervision, Funding acquisition, Writing—original draft, Writing—review and editing

## Author ORCIDs
Nicanor González-Morales (iD) http://orcid.org/0000-0003-1305-8992
Frieder Schöck (iD) https://orcid.org/0000-0002-1351-0574

## Decision letter and Author response
Decision letter https://doi.org/10.7554/eLife.50496.sa1
Author response https://doi.org/10.7554/eLife.50496.sa2

## Additional files

### Supplementary files
• Supplementary file 1. Detailed genotypes for all figures.

• Transparent reporting form

### Data availability
All data generated or analysed during this study are included in the manuscript and supporting files.

The following previously published dataset was used:

| Author(s) | Year | Dataset title | Dataset URL | Database and Identifier |
| --- | --- | --- | --- | --- |
| Spletter ML, Barz C, Yeroslaviz A, Zhang X, Lemke SB, Bonnard A, Brunner E, Cardone G, Basler K, Habermann BH, Schnorrer F | 2018 | Systematic transcriptomics reveals a biphasic mode of sarcomere morphogenesis in flight muscles regulated by Spalt | https://www.ncbi.nlm.nih.gov/geo/query/acc.cgi?acc=GSE107247 | NCBI Gene Expression Omnibus, GSE107247 |

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
