## [Decision Letter]

**Acceptance summary:**

Sarcomere formation during muscle development is a complex multi-step process that assembles a gigantic macromolecular machine, which will generate the force for future muscle contractions. Gonzalez-Morales and co-workers present a comprehensive, technically and intellectually sophisticated analysis of the function of Zasp proteins during sarcomere development using the *Drosophila* model. The authors discovered that the Z-disc associated alternatively spliced proteins (Zasps) regulate sarcomere diameter by setting the Z-disc diameter using a new mechanism of Zasp protein oligomerization: depending on presence or absence of multiple LIMdomains in the alternatively spliced Zasp isoforms, Zasp proteins can oligomerize and the Z-disc can grow to the desired diameter during muscle development. This work is a fundamental step ahead in the complex field of sarcomerogenesis and will be highly valuable to inform sarcomere development studies in other organisms and muscle types.

**Decision letter after peer review:**

Thank you for submitting your article "Myofibril size is set by a finely tuned mechanism of protein aggregation in *Drosophila*" for consideration by *eLife*. Your article has been reviewed by three peer reviewers, including Frank Schnorrer as the Reviewing Editor and Reviewer #1, and the evaluation has been overseen by Anna Akhmanova as the Senior Editor.

The reviewers have discussed the reviews with one another and the Reviewing Editor has drafted this decision to help you prepare a revised submission.

This manuscript presents a comprehensive, technically and intellectually sophisticated analysis of the members of the muscle actinin-associated LIM domain, PDZ domain family proteins, called Zasp proteins in controlling myofibril width in *Drosophila*. The authors use cellular, genetic and biochemical protein-protein interaction assays and developmental in vivo data to suggest that temporally differentially expressed long and short isoforms of the Zasp proteins control the size of the Z-disc in flight muscles. Mechanistically, the authors suggest a finely balanced interaction code of homo and heterodimers that depends on the differentially expressed LIM and ZM domains of the long, LIM carrying (growing) vs. short, devoid of LIM (blocking) isoforms.

All three reviewers agree that this work is a fundamental step ahead in the complicated field of Z-disc morphogenesis and will also be highly valuable to inform studies in other organisms and muscle types.

However, all reviewers suggest a number of essential points the authors need to address, some of which relate to the terminology used.

Essential revisions:

Terminology – The terminology is often ambiguous and confusing.

1) The term "aggregation" is unfortunate for describing macromolecular assemblies of Zasp proteins. Z-discs are very elaborate structures, which are highly regular. Hence, they are rather the opposite of an undefined aggregate.

The authors observe aggregate formation upon strong over-expression of Zasp. It is unclear whether these over-expressed wild-type Zasp proteins can be compared to pathological aggregates. The authors should use a term that clearly separates misfolded protein aggregates in myopathies (which do not trigger Z-disk formation) from the multi-molecular assemblies that may be involved in Z-disk formation, where folded domains form defined interactions between native proteins. Furthermore, it seems that these large "aggregates" only occur under over-expression and "aggregates" are *not* formed by Zasps expressed under the native promoters.

2)Title – The title should be more explicit about myofibril width, and not size, as size can be sarcomere length as well. A possible amended title could be "Myofibril diameter is set by a finely tuned mechanism of protein oligomerisation in *Drosophila*".

The same ambiguity about Zasp and sarcomere size should be avoided in the text (e.g. subsection “Z-disc formation and growth is driven by multivalent aggregation of Zasp proteins”, second paragraph).

3) Multivalent refers to a domain that has multiple sites at which interactions can form. The authors present no evidence that an individual LIM domain is multivalent – each one seems to form one distinct interaction. This term is therefore confusing.

Z-disc growth hypothesis –

4) The current statement that the ratio of long to short Zasp isoforms control Z-disc diameter is over-stated and not strongly supported by the data throughout the paper. It is clear is that over-expression of Zasp52 long destroys the fibrils and generates large ectopic 'aggregates' (Figure 1B). Overexpression of the short isoforms does not lead to aggregate formation, but is also does not strongly reduce myofibril width (rest of Figure 1).

Also Figure 6 does not fully support the authors' hypothesis. It rather appears that an imbalance of the short to long isoforms results in a larger variation of myofibril width. For example over-expression of Zasp52-PP (short) causes some fibrils to be thinner but also many to be thicker, which is against the simple hypothesis. The same appears true in the *Zasp52^MI00979^*allele, the myofibrils in the image shown (Figure 6B) are very variable in width and seem to have a problem with running straight rather than being consistently thinner, at least if the image shown in Figure 6B is representative. Furthermore, the Zasp66 Zasp67 double mutants appear to result in many destroyed fibrils (Figure 7C) rather than in consistently thicker ones.

Generally, the authors use a very strong driver, Act88F-Gal4 to overexpress the Zasp isoforms. What happens with weaker drivers, such as *Mef2*. Are aggregates formed? Do the Z-discs change?

5) It would be important to provide more molecular support that the *Zasp52^MI00979^*allele is indeed null for the long isoforms. Western Blots seem straight forward. Alternatively, the generation of clean CRISPR allele. Not being null might be a reason for the rather weak phenotype.

6) The localisations of the long vs. short Zasp variants on the Z-disc in Figure 5F-H is very impressive. However, it is not clear from the Materials and methods how these data were acquired. Have the authors done sections, e.g. using a microtome, and imaged the slices in 3D including Z-plane information? The radial distribution of Zasp isoforms could be influenced by many technical factors. How was it ascertained that the same Z-disk plane was imaged and averaged? Please explain what thickness the transverse sections were and how they were imaged.

Is it possible to image both Zasp52 long and Zasp66-GFP at the same time in these sections to have an internal control? As antibodies are available, this should be feasible.

7) Does increased or decreased Z-disc width change sarcomere length? Likely not, but it would be interesting see quantitative data.

8) Please provide a data table with all the relevant measures used to make the graphs of the paper including animal numbers etc.

---

## [Author Response]

Essential revisions:Terminology – The terminology is often ambiguous and confusing.1) The term "aggregation" is unfortunate for describing macromolecular assemblies of Zasp proteins. Z-discs are very elaborate structures, which are highly regular. Hence, they are rather the opposite of an undefined aggregate.The authors observe aggregate formation upon strong over-expression of Zasp. It is unclear whether these over-expressed wild-type Zasp proteins can be compared to pathological aggregates. The authors should use a term that clearly separates misfolded protein aggregates in myopathies (which do not trigger Z-disk formation) from the multi-molecular assemblies that may be involved in Z-disk formation, where folded domains form defined interactions between native proteins. Furthermore, it seems that these large "aggregates" only occur under over-expression and "aggregates" are not formed by Zasps expressed under the native promoters.

These are good points. We now use more specific language throughout, in particular changing sarcomere size to sarcomere diameter, and changing aggregation to oligomerization. In addition, we provide data showing that Z-disc overgrowth results in aggregate formation (Figure 1J and L), i.e., these two phenomena are linked by expression levels of Zasp52 growing isoforms: weak overexpression results in increased sarcomere diameter, strong overexpression results in aggregates.

2)Title – The title should be more explicit about myofibril width, and not size, as size can be sarcomere length as well. A possible amended title could be "Myofibril diameter is set by a finely tuned mechanism of protein oligomerisation in Drosophila".The same ambiguity about Zasp and sarcomere size should be avoided in the text (e.g. subsection “Z-disc formation and growth is driven by multivalent aggregation of Zasp proteins”, second paragraph).

We changed the title accordingly.

3) Multivalent refers to a domain that has multiple sites at which interactions can form. The authors present no evidence that an individual LIM domain is multivalent – each one seems to form one distinct interaction. This term is therefore confusing.

We meant Zasp growing isoforms are multivalent, they have multiple interaction sites (multiple, meaning more than one LIM domain). We adjusted the wording.

Z-disc growth hypothesis –4) The current statement that the ratio of long to short Zasp isoforms control Z-disc diameter is over-stated and not strongly supported by the data throughout the paper. It is clear is that over-expression of Zasp52 long destroys the fibrils and generates large ectopic 'aggregates' (Figure 1B). Overexpression of the short isoforms does not lead to aggregate formation, but is also does not strongly reduce myofibril width (rest of Figure 1).Also Figure 6 does not fully support the authors' hypothesis. It rather appears that an imbalance of the short to long isoforms results in a larger variation of myofibril width. For example over-expression of Zasp52-PP (short) causes some fibrils to be thinner but also many to be thicker, which is against the simple hypothesis. The same appears true in the Zasp52^MI00979^ allele, the myofibrils in the image shown (Figure 6B) are very variable in width and seem to have a problem with running straight rather than being consistently thinner, at least if the image shown in Figure 6B is representative. Furthermore, the Zasp66 Zasp67 double mutants appear to result in many destroyed fibrils (Figure 7C) rather than in consistently thicker ones.Generally, the authors use a very strong driver, Act88F-Gal4 to overexpress the Zasp isoforms. What happens with weaker drivers, such as Mef2. Are aggregates formed? Do the Z-discs change?

We find that *Mef2*-Gal4 is a strong driver, similar to Act88F (and resulting in similar aggregates, data not shown). To test a weaker overexpression we used UH3-Gal4 that is a much weaker driver than Act88F and has a similar temporal expression profile.

Overexpression of Zasp52-PR using UH3-Gal4 does not form aggregates and instead results in bigger than normal Z-discs. We have added these results to Figure 1J-L.

*Zasp52^MI00979^* mutants have predominantly smaller Z-disc, with only a minimal increase in larger Z-discs (see quantification in Figure 6C). In contrast, *Zasp52^MI02988^*, which affects the small isoforms of Zasp52 have an increase in the proportion of big Z-discs, but not small ones. These two observations are consistent with our model.

The overexpression data is harder to fit into the model. As the reviewers pointed out, overexpression of blocking isoforms leads to a larger variation in Z-disc diameter rather than just an increase in small Z-discs. The important observation is that overexpression of any of the three blocking isoforms results in smaller discs and that this can be reverted if the ZM domain is mutated. Bigger Z-discs might result from the forced addition of Zasp proteins into the Z-disc independently of their role as blocking isoforms. Overexpression data are always somewhat artificial and not necessarily the exact opposite of the mutation. The Zasp66 and Zasp67 mutant phenotypes presented in Figure 7A-E are undoubtedly more convincing and confirm our model.

5) It would be important to provide more molecular support that the Zasp52^MI00979^ allele is indeed null for the long isoforms. Western Blots seem straight forward. Alternatively, the generation of clean CRISPR allele. Not being null might be a reason for the rather weak phenotype.

We have already characterized the depletion of the long isoforms by western blotting in the Zasp^MI00979^ allele in Liao K., et al., 2016. Figure 6B). There appears to be a small amount of readthrough (Figure 6C), visible upon long exposure and increased loading of gel, indicating that all 3 alleles (02988, 07547, 00979) are strong hypomorphs.

6) The localisations of the long vs. short Zasp variants on the Z-disc in Figure 5F-H is very impressive. However, it is not clear from the Materials and methods how these data were acquired. Have the authors done sections, e.g. using a microtome, and imaged the slices in 3D including Z-plane information? The radial distribution of Zasp isoforms could be influenced by many technical factors. How was it ascertained that the same Z-disk plane was imaged and averaged? Please explain what thickness the transverse sections were and how they were imaged.Is it possible to image both Zasp52 long and Zasp66-GFP at the same time in these sections to have an internal control? As antibodies are available, this should be feasible.

We used phalloidin staining as an internal control and we now include the counterstains as suggested (revised Figure 5). To detect specifically the long isoforms of Zasp52 we used a GFP protein trap allele [Zasp52-ZCL]. We have several antibodies for Zasp52 but they all recognize the N-terminal region which mostly labels the short isoforms. To image Zasp66 or Zasp67 we only have GFP trap alleles. Therefore, we cannot image the long isoforms of Zasp52 and Zasp66 at the same time.

We also added more details to the Materials and methods section and a supplementary figure to better explain how the images were obtained (Figure 5—figure supplement 1).

7) Does increased or decreased Z-disc width change sarcomere length? Likely not, but it would be interesting see quantitative data.

*Zasp52^MI00979^* mutants are slightly shorter than the control. We don’t have a clear explanation for this, sarcomere length regulation is a very complex matter and at this point it is unclear if Zasp plays a role or if these defects are indirect effects of perturbing Z-disc composition. We prefer not to include this data in the main text.

8) Please provide a data table with all the relevant measures used to make the graphs of the paper including animal numbers etc.

We have included excel files with all measurements (Source Data files).